# CommsVAE: Learning the brain's macroscale communication dynamics using coupled sequential VAEs

## Abstract

Communication within or between complex systems is commonplace in the natural sciences and fields such as graph neural networks. The brain is a perfect example of such a complex system, where communication between brain regions is constantly being orchestrated. To analyze communication, the brain is often split up into anatomical regions that each perform certain computations. These regions must interact and communicate with each other to perform tasks and support higher-level cognition. On a macroscale, these regions communicate through signal propagation along the cortex and along white matter tracts over longer distances. When and what types of signals are communicated over time is an unsolved problem and is often studied using either functional or structural data. In this paper, we propose a non-linear generative approach to communication from functional data. We address three issues with common connectivity approaches by explicitly modeling the directionality of communication, finding communication at each timestep, and encouraging sparsity. To evaluate our model, we simulate temporal data that has sparse communication between nodes embedded in it and show that our model can uncover the expected communication dynamics. Subsequently, we apply our model to temporal neural data from multiple tasks and show that our approach models communication that is more specific to each task. The specificity of our method means it can have an impact on the understanding of psychiatric disorders, which are believed to be related to highly specific communication between brain regions compared to controls. In sum, we propose a general model for dynamic communication learning on graphs, and show its applicability to a subfield of the natural sciences, with potential widespread scientific impact.

## 1 Introduction

Characterizing macroscale communication between brain regions is a complex and difficult problem, but is necessary to understand the connection between brain activity and behavior. The effect of neural systems on each other, or connectivity, has also been linked to psychiatric disorders, such as schizophrenia Friston (2002). Hence, gaining a deeper understanding of the dynamics underlying the communication between brain regions is both from the perspective of understanding how our brain facilitates higher-order cognition and also to provide insight into and consequently how psychiatric disorders arise. Static functional network connectivity (sFNC) and dynamic functional network connectivity (dFNC), computed respectively as the Pearson correlation between regional activation timeseries over the full scan duration (sFNC) or on shorter sliding windows (dFNC), are among the most widely reported measures of connectivity between brain regions Hutchison et al. (2013). These approaches calculate the correlation between the timeseries of each brain region to find their coherence, either across the full timeseries (sFNC) or by windowing the timeseries and calculating the correlation within each window (dFNC). Although extensions have been proposed Hutchison et al. (2013), along with more complex connectivity measures, such as wavelet coherence Yaesoubi et al. (2015), multiplication of temporal derivates Shine et al. (2015), and Granger causality Roebroeck et al. (2005); Seth et al. (2015), Pearson correlation remains the most prevalent measure of brain network connectivity. And even the less commonly employed metrics have issues stemming from some combination of sensitivity to noise, linearity, symmetry, or coarse timescales. Most im-

portantly, these approaches do not directly model the communication, but rather analyze it post-hoc. The pursuit of instantaneous communication between brain regions Sporns et al. (2021) and generative approaches to model communication can potentially lead to models that closely resemble effective connectivity Avena-Koenigsberger et al. (2018). An important advantage of generative models is that they allow us to move away from post-hoc inference from context-naïve metrics toward simulating macroscale brain communication. We propose the use of recurrent neural networks as generative models of communication on both simulated data and functional magnetic resonance imaging (fMRI) data to validate and demonstrate the specificity of our model. Our method complements connectivity metrics, since it does more than quantify the aggregation of communication Avena-Koenigsberger et al. (2018), but directly simulates it, and can thus be analyzed using those same connectivity metrics.

Creating an accurate generative model of the macroscale communication dynamics in the brain is hard, due to its complexity. However, there are some general design principles the brain follows Sterling & Laughlin (2015). Generally, the brain tries to minimize its energy use, which is likely also true for communication in the brain. Macroscale communication is an energy-intensive process and involves white matter tracts, which are essentially highways that connect spatially separate parts of the brain. Hence, the amount of information and the number of times information is sent should generally be limited. The bits needed to convey are limited by the brain using sparse coding, which at a lower scale is how neurons encode and communicate information Olshausen & Field (2004). Although mechanistically different, due to metabolic and volume constraints, macroscale communication presumably exhibits strategies similar to sparse coding to efficiently transfer information between neural populations Bullmore & Sporns (2012); Sterling & Laughlin (2015). To incorporate this inductive bias into our model, we regularize the communication from one region in our model to another using a KL-divergence term to a Laplace distribution. The communication itself is modeled as a Laplace distribution and by minimizing its divergence from a Laplace distribution we encourage sparser temporal communication. Encouraging sparse temporal communication implies that information is only sent when necessary, and lower rates of communication lead to reductions in the brain's energy requirements.

As far as we are aware, this is the first model that explicitly models communication dynamics between vertices on a graph, specifically between brain regions. The connectivity metrics that are used currently assume that the connectivity between brain regions is stationary, e.g. Granger causality, and lack a generative model. Although non-linear generative models have been proposed for connectivity Stephan et al. (2008), however, determining the correct model is intractable Lohmann et al. (2012). Furthermore, most methods quantify the coupling between brain regions and do not consider potentially rapid changes in fMRI signals that can be traced from region to region. In this paper, we specifically try to model the communication between brain regions, which likely relates to these abrupt and parsimonious changes in the signal. This means that our model is finding a type of interaction between brain regions, or more generally nodes on a graph, that has never been studied in this way before, to the best of our knowledge. The assumptions we make thus also differ from commonly used connectivity metrics. Firstly, we assume directionality and communication with the same temporal resolution as the original signal. Hence, we completely move away from windowed approaches, that quantify the coupling between brain regions within windows. Secondly, we assume that brain regions are largely independent and can be modeled by a dynamical system with sparse inputs. Lastly, we aim to learn a fully generative model of fMRI data, where the communications and initial state of each brain region's dynamical system can be sampled from a simple distribution. This also implies that our method is hard to compare with more common connectivity metrics, and we expect it to exhibit different behavior since it is modeled under different assumptions. We propose this model because we believe it has tremendous value and complements many connectivity metrics in a meaningful way, not only in the neuroimaging community, but also in other scientific fields, such as sociology Barnes (1969), computational biology or chemistry Balaban (1985), and traffic prediction models Zhao et al. (2019).

The goal of our model is to get a better idea of communication dynamics in the brain. To evaluate whether our model is equipped to find the underlying dynamics of a known generative model, we first train it on simulated data. After we show that our model finds the correct generative model from the simulated data, we apply our model to neuroimaging data with fairly well-established neural pathways. Although the pathways are well-established, we do not know the exact ground truth communication underlying fMRI data. We do know that communication dynamics depend on

the task it is trying to perform, even outside of explicit windows in which these tasks are performed. Hence, to evaluate our model on fMRI data, we train it on three different tasks and use logistic regression to predict what task a small window of timesteps is from using the communication our model predicts. We compare our model to dFNC and show that our method is more sensitive to the task, especially for small windows where no explicit task is performed. This has implications for both resting-state fMRI and task fMRI, since our model may be able to provide a deeper look into how tasks shape macroscale communications in the brain. Understanding task responses allows us to more accurately understand their correlates to behavior and how communication in the brain shapes certain behavior. Since our results indicate that our model is more sensitive to communications even at resting-state that are still related to a task. Being sensitive even outside of external driving of fMRI activity is impactful for resting-state fMRI and psychiatric disorder research, where we expect communication dynamics to be different between controls and patients. Furthermore, since we establish our method on simulated data and then extend it to real data, we provide an initial framework for scientists that may want to extend our method into their area of expertise.

## 2  METHOD

**Formal definition of our model**  The data structure we assume in this work is a simple graph $\mathcal{G} = (V, E)$, with $N$ vertices ($V$), and $M$ edges ($E$), as shown in Figure 1A. Generally, we can assume a fully-connected graph, since our proposed model can learn to not send any communication over an edge, meaning it essentially 'trims' that edge. Each vertex $v^i$ has a corresponding timeseries $\mathbf{x}^i_{1\ldots T}$ with $T$ timesteps, this timeseries is modeled as the following dynamical system.

$$\dot{\mathbf{x}}^i = \mathbf{F}^i(\mathbf{x}^i(t), \mathbf{u}^i(t)) \tag{1}$$

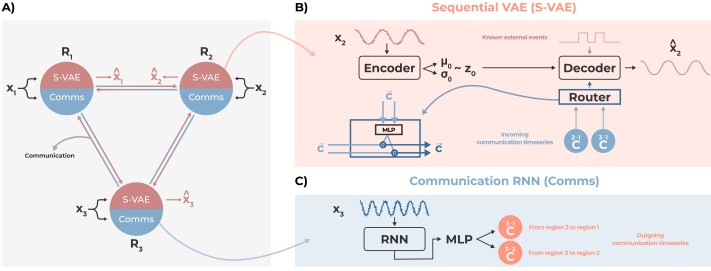

Figure 1: This figure shows the composition of a single node from a graph on the left. The communication and reconstruction RNNs are completely independent of each other, the communication RNN sends signals to other nodes, whereas the reconstruction RNN reconstructs the original timeseries from outside communication and an initial condition $z_0$ (right).

One recent model that has been used to model this dynamical system is LFADs  Pandarinath et al. (2018), where they model the external inputs $\mathbf{u}(t)$ as samples from a normal distribution parameterized by a controller that has access to the hidden state of the encoder and decoder. The controller can be fairly opaque in what it is concretely modeling, however, if we have access to timeseries for multiple nodes, we can replace the input from the controller with input coming from the neighbors of a node. This incorporates a more explicit inductive bias in our model. Formally, the external inputs for vertex $v^i$ are communications $c^{\rightarrow i}_{1\ldots T}$ and external events $\mathbf{e}(t)$. We use the notation $\rightarrow i$ to mean all incoming communication from neighbors of vertex $v^i$, $i \rightarrow$ to mean all outgoing communication from vertex $v^i$ to its neighbors, and $j \rightarrow i$ to mean communication from vertex $v^j$ specifically to vertex $v^i$. The external events are known external perturbations of the dynamical system, such as the block design of a task. To model the dynamical system of each vertex separately, we train a recurrent neural network (RNN), which we call the decoder, to find the non-linear function ($\mathbf{F}^i_\theta$) that describes its dynamics. First, we consider the case where the model receives no inputs, this is a transient dynamical system, that only depends on its initial state $\mathbf{z}^i_0$. We find the initial inputs to the dynamical systems using an encoder, with the following probability distribution $p(\mathbf{z}^i|\mathbf{x}^i_{1\ldots T}) = \mathcal{N}\left(\mathbf{z}^i; \boldsymbol{\mu}^i_0, \boldsymbol{\sigma}^i_0\right)$. This distribution is parameterized using a bi-directional RNN (the encoder), and we sample the initial state from that distribution $\mathbf{z}_0 \sim p(\mathbf{z}^i|\mathbf{x}^i_{1\ldots T})$. The encoder and

decoder together form a sequential variational autoencoder (S-VAE) Zhao et al. (2017); Kingma & Welling (2013). The S-VAE is visualized in Figure 1B. The decoder can be expressed as parameterizing the following distribution $p(\hat{x}^i_{1...T}|\mathbf{z}^i_0)$, which allows us to train it using the evidence lower bound (ELBO) Kingma & Welling (2013).

We extend this model by adding two types of inputs, namely the communication from other vertices, and external events. The communication from one vertex to another is also modeled as a recurrent neural network, with $\mathbf{c}^i$ as a sample from a Laplace distribution. $p(\mathbf{c}^{i\ toj}_t|\mathbf{x}^i_{1...t-1}) = \mathcal{L}\left(\mathbf{c}^{i\rightarrow j}_t; \boldsymbol{\mu}^{i\rightarrow j}_t, \boldsymbol{\sigma}^{i\rightarrow j}_t\right)$. Communication from vertex $i$ to vertex $j$ at each timestep $t$ is thus sampled from the following Laplace distribution $\mathbf{c}^{i\rightarrow j}_t \sim p(\mathbf{c}^{i\rightarrow j}_t|\mathbf{x}^i_{1...t})$. Modeling the communication timeseries between vertices as distributions for each timestep allows us to regularize the communication. To encourage sparse communication between brain regions, we minimize the KL-divergence between the communication timeseries and a zero-mean unit-variance multi-dimensional Laplace distribution during training. The communication network is visualized in Figure 1C. Note that communication from vertex $i$ to vertex $j$ computed at time $t$ arrives at the decoder at time $t + 1$ to ensure the model is temporally causal. Thus, the decoder now models the following distribution $p(\hat{x}^i_{1...T}|\mathbf{z}^i_0, \mathbf{c}^{\rightarrow i}_{1...T-1}, \mathbf{e}_{1...T})$, where $\mathbf{e}_{1...T}$ are the external events. In our model, we assume that external events only affect one brain region, after which the signal propagates to other brain regions through communication. The external events are represented as one when the external event is active and zero when it is inactive. Our model trains a vector for each event type. Each entry in the vector corresponds to the probability that an event should be inserted as an external event in that brain region. The vector parameterizes a Gumbel-Softmax distribution Jang et al. (2016), which we sample from to determine the index of the brain region that gets the external event as input.

So far, each communication network predicts communication to another brain region independently. To allow the model to suppress information that has already been sent by other brain regions, we add a router network that is similar to the forget gate in a long short-term memory (LSTM) network Hochreiter & Schmidhuber (1997). The router network receives all the incoming communication $\mathbf{c}^{\rightarrow i}_t$ at timestep $t$ and this vector is fed through a multi-layer perceptron (MLP), which has the same number of outputs as the size of the communication vector $\mathbf{c}^{\rightarrow i}_t$. The output is transformed through a sigmoid to be between [0, 1] and multiplied with the incoming communication vector. This allows the router to block information sent from a certain vertex that is already captured by information sent from other vertices. The router is visually represented in Figure 1A.

With simplified notation, we thus have the following equations for the S-VAE.

$$\mathbf{h}^i_T = \text{RNN}^i_{\text{Enc}}(\mathbf{x}^i_{1...T}) \tag{2}$$

$$\boldsymbol{\mu}^i_0 = \mathbf{W}^i_\mu \mathbf{h}^i_T, \qquad \boldsymbol{\sigma}^i_0 = \mathbf{W}^i_\sigma \mathbf{h}^i_T, \qquad \mathbf{z}^i_0 \sim \mathcal{N}\left(\boldsymbol{\mu}^i_0, \boldsymbol{\sigma}^i_0\right) \tag{3}$$

$$\hat{\mathbf{x}}^i_{1...T} = \text{RNN}^i_{\text{Dec}}(\mathbf{z}^i_0, \mathbf{c}^{\rightarrow i}_{1...T}, \mathbf{e}_{1...T}) \tag{4}$$

Then, for the communication network we obtain the following equations.

$$\mathbf{h}^i_t = \text{RNN}^i_{\text{Comm}}(\mathbf{x}^i_{\leq t}) \tag{5}$$

$$\boldsymbol{\mu}^{i\rightarrow}_t = \text{MLP}^i_{\text{Comm},\mu}(\mathbf{h}^i_{t-1}), \qquad \boldsymbol{\sigma}^{i\rightarrow}_t = \text{MLP}^i_{\text{Comm},\sigma}(\mathbf{h}^i_{t-1}), \qquad \mathbf{c}^{i\rightarrow}_t \sim \mathcal{L}\left(\boldsymbol{\mu}^{i\rightarrow}_t, \boldsymbol{\sigma}^{i\rightarrow}_t\right) \tag{6}$$

$$\mathbf{c}^{i\rightarrow}_t = \mathbf{c}^{\rightarrow i}_t \odot \sigma\left(\text{MLP}^i_{\text{Router}}\left(\mathbf{c}^{\rightarrow i}_t\right)\right) \text{‘} \tag{7}$$

Where $\sigma(\cdot)$ is a sigmoid function, and $\text{MLP}^i_{\text{Comm},\mu}$ and $\text{MLP}^i_{\text{Comm},\sigma}$ share every layer, except the last one.

We can then adapt the evidence lower-bound (ELBO) Kingma & Welling (2013) as follows, where $q^i_\phi\left(\mathbf{z}^i_0, \mathbf{c}^{i\rightarrow}_{1...T} \mid \mathbf{x}^i_{1...T}\right)$ refers to the approximate posterior, and $p^i_\theta\left(\mathbf{x}^i_{1...T} \mid \mathbf{z}^i_0, \mathbf{c}^{\rightarrow i}_{1...T}, \mathbf{e}_{1...T}\right)$ is the generative network, in our case the recurrent decoder.

$$\begin{aligned}
\mathcal{L}(\theta, \phi, \mathbf{x}_{1...T}) := \sum_{i=1}^{N} \frac{1}{N} \Big[ &- \text{D}_{\text{KL}}\left(q_\phi(\mathbf{z}^i_0 \mid \mathbf{x}^i_{1...T}) \| p(\mathbf{z})\right) \\
&- \lambda \text{D}_{\text{KL}}\left(q^i_\phi(\mathbf{c}^{i\rightarrow}_{1...T} \mid \mathbf{x}^i_{1...T}) \| p(\mathbf{c})\right) \\
&+ \mathbf{E}_{q^i_\phi\left(\mathbf{z}^i_0, \mathbf{c}^{i\rightarrow}_{1...T}|\mathbf{x}^i_{1...T}\right)} \left[\log p^i_\theta\left(\mathbf{x}^i_{1...T} \mid \mathbf{z}^i_0, \mathbf{c}^{\rightarrow i}_{1...T}, \mathbf{e}_{1...T}\right)\right] \Big]
\end{aligned} \tag{8}$$

Where $p(\mathbf{z})$ and $p(\mathbf{c})$ are the priors for the initial inputs and the communications and $\lambda$ is the sparsity multiplier, respectively. Specifically, the first row of the equation 8 refers to the KL-divergence between the initial hidden states $\mathbf{z}_0^i$, learned by the encoder ($q_\phi^i$) and the prior $p(\mathbf{z})$, which is a zero-mean one standard deviation Gaussian. The second row of the equation refers to the KL-divergence, weighted by a sparsity parameter $\lambda$, between the communication timeseries ($\mathbf{c}_{1,\dots,T}^{i\rightarrow}$) and a prior $p(\mathbf{c})$, which is a zero-mean $0.1$ standard deviation Laplace distribution. This encourages the model to learn sparse communication timeseries between the vertices. The final row of the equation refers to the expected log-likelihood, or the reconstruction, of the signal that is modeled by the decoder ($p_\theta^i$). We assume that the fMRI signal for each region is normally distributed with a constant standard deviation, which means that optimizing the expected log-likelihood is equivalent to minimizing the mean-squared error. In our work, we decide to minimize the mean-squared error because we found it to be more stable.

**Intuition and synthetic data**  To gain some intuition about our model, we explain how our model works on the simulated data we use in this work. We assume that if part of the signal is explained by a pattern in another region at a previous timestep, communication could have occurred. A simple example is communication between two regions, where the timeseries for each region is uniform noise. Let us assume there is a linearly decaying pulse in the first region at a time $T1 \sim \text{Uniform}(0, 70)$, that decays until $T2 \sim T1 + 20$. Once the pulse has decayed in the first region at time $T2$, a pulse starts in the second region that linearly decays at time $T3 \sim T2 + 20$. These are exactly the parameters we use to create our synthetic data. Now, if we only observe one of the two timeseries, the pulses will seem sampled from a uniform distribution to the decoder of that region. However, if we have access to both regions at the same time and enough training data, our model should be able to figure out that the pulse in the first region structurally precedes the pulse in the second region. In our synthetic data, we add a third region that models a sinusoid and has a pulse after the second region's pulse that linearly decays at time $T4\ T3 + 20$. This region also has a different underlying signal, namely a randomly shifted sinusoid (shifted between Uniform$(-20, 20)$) to show that our model can separate global signal from communication. A visual example of our synthetic data is shown in Figure 2A. We expect that the initial inputs $z_0$ models the global signal so that the communication our model predicts is not affected by the sinusoidal signal. As mentioned previously, fMRI data also exhibits global signals that may affect connectivity metrics, such as physiological noise. Hence, we can infer whether our model is affected by the global signal in our simulation data to make it more realistic when transferred to biological data. The communication we thus expect is shown in Figure 2B. Namely, the communication model in region 1 should learn to send a signal to region 2 when the pulse in its region has almost decayed, and region 2's communication model should learn to send a pulse to region 3 once its pulse has almost decayed. The expected communications in our synthetic data are shown in Figure 2B.

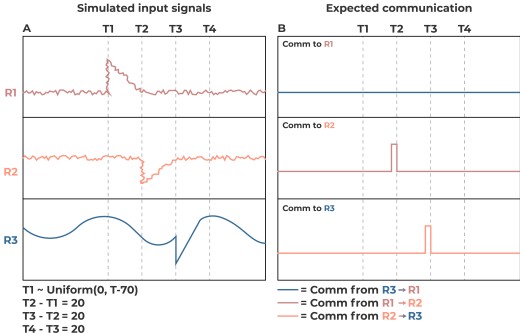

Figure 2: This figure shows the simulated input signals on the left and the communication we would expect our model to find. Both signals are drawn using graphical design software. The communication would happen from R1 to R2 (middle right) right before T2, and from R2 to R3 (bottom right) right before T3.

**Biological data**  The fMRI data we use in this work is obtained from the open-access, under data usage terms, HCP-1200 dataset  Van Essen et al. (2013) for all subjects with cortical surface time-

series data. We use resting-state fMRI (rs-fMRI) and three tasks in our work, the motor, emotion processing, and language processing tasks. The data is registered using multimodal surface registration (MSM) Robinson et al. (2014; 2018), and surfaces are constructed using Freesurfer Glasser et al. (2013); Fischl (2012). We use the Yeo-17 atlas Yeo et al. (2011) to separate the cortical surface data into 17 separate regions. The spatial information within each region is averaged, as is commonly done in communication analyses of the brain. Each subject's timeseries is then independently band-pass filtered independently ($0.01 - 0.15$Hz) and linearly detrended using the Nilearn package Abraham et al. (2014). Each task has a different length, so we cut off tasks at 176 timesteps, which is the smallest common size of the timeseries. For the rs-fMRI data we use the first 200 timesteps to make the length similar to the task data. The motor task consists of a visual cue, after which a subject has to start performing a motor task. The motor tasks a subject performs are related to the left hand, right hand, left foot, right foot, and tongue. For the emotion task, the subject sees a visual cue and then has to match two shapes shown at the bottom of the screen to a shape at the top. This sub-task is referred to as the neutral task, and the other task has the subject match one of two faces at the bottom of the screen with a face shown at the top. The faces either look fearful or angry. The language task consists of a sub-task where the participant has to perform arithmetic and push a button for the right answer. The story sub-task presents the subjects with a short story of 5-9 sentences. The timing of each task is shown, after convolving it with SPM's hemodynamic response function Penny et al. (2011), in Appendix B. We train one model on all three tasks such that the prediction of the task is not influenced by the fact that the model was specifically trained to model communication for that task.

**Implementation**   Our model is implemented in Pytorch Paszke et al. (2017) and trained on an internal cluster using single NVIDIA V100 and NVIDIA 2080 GPUs, with a batch size of $8$, the Adam optimizer Kingma & Ba (2014), a 1E-4 weight decay, a learning rate of 0.01, 0.1 epsilon, and 0.9, 0.999 as betas. We reduce the learning rate when it plateaus using a scheduler. At each plateau, we reduce the learning rate by multiplying it with 0.95, with a patience of 6 epochs, and a minimum learning rate of $1E - 5$. We train each model for 200 epochs and perform early stopping when the loss has not improved for 50 epochs, for the simulated data across 8 seeds: (42, 1337, 9999, 50, 100, 500, 1000, 5000). The model is trained on the first three seeds for the biological data, this is due to computational limitations. Since we use multiple sparsities, the number of runs quickly explodes, and each model on biological data takes about $7 - 8 : 30$ hours to train. All of the code will be made publicly available after the double-blind review has concluded on GitHub. The hidden size we use for the encoder, decoder, and communication model are all $16$. For the synthetic data we use sparsities: (0.001, 0.005, 0.01, 0.05, 0.1, 0.5, 1.0, 2.0). For the real data we use a smaller number of sparsities due to computation limitations, but also because we have narrowed the sparsity down at this point to: (0.001, 0.005, 0.01, 0.05, 0.1, 1.0). To not mix subjects across the training, validation, and test set, we split them based on the subjects. Hence, a subject occurs with every task in one of the sets it was assigned to and in no other sets. Data is z-scored across the temporal dimension of the data for each subject independently.

**Experiments**   To test whether our model is interesting as a complement to connectivity metrics, we evaluate our model in three different ways. To get a better idea of how much the controller improves our model, we perform each experiment with both models. We refer to the model without the controller as CommsVAE and the model with the controller as Meta-CommsVAE in the rest of the text. Firstly, we evaluate our model on the aforementioned synthetic data. We calculate the correspondence between the expected communications and the communication that our model predicts on a test set and look at how the sparsity parameter affects the average correspondence to the test set. The correspondence is calculated by convolving a normal distribution over an impulse response centered on the start of the event in that brain region. Then we correlate the communication between the resulting signal and the communication to that brain region for different shifts of the communication signal. Finally, we sum the maximum absolute correlations for $R1- > R2$ and $R2- > R3$, and subtract the sum of two times the average absolute communication for all other brain regions. This corresponds to expecting zero communication in all other brain regions, and a spike right before the start of the actual event. By using shifts to calculate the correlation we do not make assumptions about the lag that is learned between the communication RNN and the decoder. Note that when we use communication in these experiments, we take the mean of the Gaussian or Laplace distribution, since it is the most probable sample under that distribution. Then, we train both our models on an fMRI dataset with three tasks, each model is trained on all three tasks

conjointly not to induce any task-specific model noise that can affect the downstream prediction. Since we do not have a ground truth for the communication of the fMRI signal, we determine the best model by looking at the reconstruction error on the validation set with respect to the sparsity, see Appendix A. We use the model that performs the best across the seeds and use it for task prediction. We use 4 different window sizes [5, 10, 20, and 50] to compare our model to dFNC, Granger causality Granger (1969), transfer entropy Lobier et al. (2014), multiplication temporal derivatives Shine et al. (2015), and the full fMRI signal in the window for each region. This allows us to determine whether our model has better temporal resolution since the quality of the correlation will likely get lower with a smaller dNFC window. For each window, we take the absolute value of our model's communication (since it can be both negative and positive) and sum it across each window. Then, we train a logistic regression classifier on the training and validation set using the temporally summed communication to predict what task each subject in the test set's window is from. This results in a classification accuracy across time for the four windows, and we perform the same routine for the connectivity metric comparisons. Finally, we look at the absolute correlation between the communication timeseries for each directed edge and the sub-tasks, we average the correlation across seeds and the subjects in the test set. We visualize the averaged correlations that are higher than 80% of the maximum value or higher than 0.2, whichever value is higher. This experiment allows us to see whether our method sends biologically realistic compared to a ground-truth.

## 3 RESULTS

**Synthetic data**   The results for the synthetic data are shown in Figure 3. Generally, we see a trend where higher sparsity parameters increase the correspondence to our expected communication on the test set, up to a point where increased sparsity suppresses any communication. There is a peak around 0.01 in Figure 3A for both models, after which the correspondence with the expected communication decreases. This exactly supports our hypothesis that encouraging sparse communication helps. To show what the communication looks like in the two models we test, we take the test set synthetic data, predict the communication for each, and use the synthetic generation parameters to align the communication timeseries. After aligning the communication timeseries, we take the average over the test set for the best seeds for each model at 0.01 sparsity from Figure 3A and visualize what the communication looks like in Figure 3B. Right before the pulse ends in the region 1, it communicates with region 2 before its pulse begins. The communication from region 2 to region 3 starts exactly when the pulse is almost linearly decayed, right before the pulse starts in region 3. The main difference between our Meta-CommsVAE and CommsVAE model is that in Figure 3B, the CommsVAE's communications between region 1 and region 2 are closer to the actual pulse. Generally, we do not see a trend in terms of a difference between the two CommsVAE models, at least for the synthetic data. This is probably because the task itself is sparse and hence being able to increase sparsity using the controller is unnecessary. An example of the reconstruction of validation data by our model is shown in Appendix C.

3

**fMRI data**   The results to test both the specificity of our generative model and how instantaneous the communication is are shown in Figure 4. We produce these results for the best model we find based on the reconstruction error on the validation set, see Appendix A. The best model is the Meta-CommsVAE model with a sparsity of 0.05. However, the standard deviation over the reconstruction is higher than for 0.01, and since the difference in the average is negligible between the two sparsities, we select 0.01 for visualizations and the next two results because it seems more stable. Our results with this model clearly show that our method outperforms all of the connectivity metrics we compare it to, and even the fMRI data itself. This indicates that our model finds representations of the tasks in the rs-fMRI data that are intuitive and not linearly separable initially. This is true for every window, and every window size. Note that both Granger causality and transfer entropy perform so poorly in terms of classification accuracy, is likely because they are inaccurate when fit in small windows. However, our method only has a small decline in classification accuracy for smaller window sizes, which supports our hypothesis that our model can find instantaneous communication. Our method already starts being task-specific early on in time, meaning that even before any of the tasks start, the communication our model finds in the brain is task-specific, which is likely to be true if we knew the ground truth of communication. One reason that task accuracy increases over time

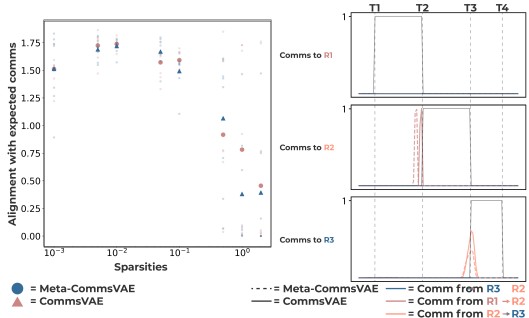

Figure 3: This figure shows the performance of our models for different sparsity parameters (x-axis) on the left and a comparison of our model's communications to the ground truth event times (black) on the right. The smaller, slightly transparent, dots in the left plot are the performances of different seeds for each model, and the larger bold dot is the average performance of the seeds. The values at which events happen are uniformly sampled, so it is hard to visualize when the model communicates since the communication is different for every test trial due to differences in event times. Hence, to average the communication over the test set, we use the event times to shift the timeseries so they align. The right figure shows the average over these aligned timeseries for the test set.

is that the communication model only has access to points previously, so early on in the timeseries, it does not have enough information to start sending high-quality communication. After a few windows, the accuracy of the task increases and is then relatively stable. This is different for the dFNC, which fluctuates, likely based on when the tasks start, but our model is specific to the task and essentially the same across all of the windows in the timeseries. Thus, our model is even task-specific when there is relatively little task signal from either of the tasks in a certain window. Another result we can gather from Figure 4 is that the lower-triangular part of the matrix is worse at predicting the task than the upper-triangular part of the communication matrix for each window. The reason we include these results is that the dFNC is symmetric and the lower and upper triangular parts of the matrix are thus the same. To make the comparison fair, we compare separately compare the triangular parts of our communication matrix with dFNC. Since our method is directed, we can also take the whole matrix, which outperforms both the lower and upper triangular parts of that matrix, as expected.

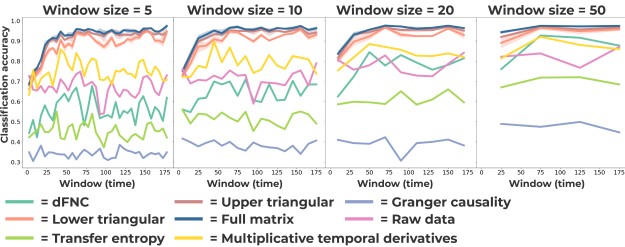

Figure 4: This figure shows four panels, each with a different window size to compute both our model's communication density and dFNC. The y-axis shows the classification accuracy of the task given only that window on the test set.

Our last experiment aims to evaluate whether our model communication between regions that aligns with the sub-tasks in the data. We use naming conventions for these regions from Kahali et al. (2021). The hemodynamic response model over time for each sub-task is shown in Appendix B. The average correlation across subject and seeds is shown in each of the sub-tasks, and color of the title indicates whether it is a sub-task of the motor task (blue), emotion task (red), or language task (orange). Two of the most easily verifiable correlations to sub-tasks are the visual cues in the bottom left of Figure 5. The edges from the early visual area (V-A) to somatomotor B (SM-B) and limbic A (L-A) are correlated to the hemodynamic response of the visual cue. This is a convincing result because of the nature of the task, namely motor execution after the visual cue. Motor execution has

been shown to be related to both the somatomotor region as well as the limbic region Mogenson et al. (1980). Although the latter is relatively understudied, our results indicate a potentially fruitful direction when related to communication. Furthermore, the visual cue for the emotion task is related to communication from the early visual area (V-A) to control A (C-A), which is a cognitive control area. Given that the emotion task requires the participant to match a shape (neutral) or a face (fear) to a target in the top of a screen, the communication from the visual area to the cognitive control area also aligns with biological intuition. Other notable communication occurs towards two default mode networks (D-A, D-B). This aligns with previous research, which indicates sub-networks of the default mode network are coupled with language networks Gordon et al. (2020).

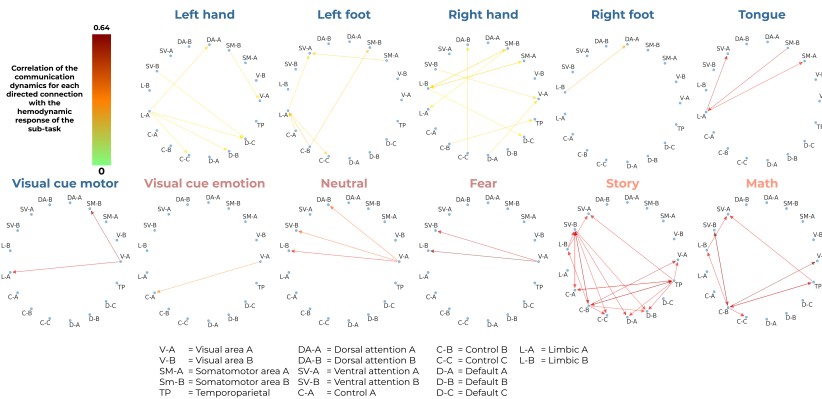

Figure 5: This figure shows the correlation between communication from our model and sub-tasks in the data. Notably, the visual cue in both the motor and emotion task induce sending from the early visual areas (V-A) to networks implicated in motor execution.

## 4  DISCUSSION

This work proposes a generative model for macroscale communication in the brain that we show predicts directed, instantaneous, and task-specific communication dynamics. We compare our model on a synthetic task that tests the requirements we set out for our model and find that our model can accurately uncover the communication dynamics we incorporate into the synthetic data. Then, we train our two models on three fMRI tasks and find that reconstruction error decreases with higher sparsity parameters, indicating that the sparse prior on the brain communication dynamics fits our data. The sparsity of macroscale brain communication needs to be studied further but aligns with findings in computational neuroscience Olshausen & Field (2004) and the design principles of the brain Sterling & Laughlin (2015). We use the best model based on reconstruction error over seeds to predict what task small time windows are from in a test set and compare the classification accuracy with a multiple commonly used connectivity metrics, both directed, and undirected. The performance of our model does not decrease as drastically with smaller window sizes, supporting our claim that our model predicts instantaneous communication. Then, to understand how communication is related to sub-tasks, we calculate the correlation between each sub-task and the communication timeseries of each edge. After averaging over test subjects, and seeds, we visualize the highest sub-task correlations. Almost all of the strong connections we find are directed, with correlations up to $0.64$. The directions and the strength of the communication also align with neuroscientific priors we have about how regions affect each other. Thus, our results confirm what we set out to incorporate into our model from the beginning.

We want to expand on this model by applying it to resting-state fMRI data, and explore whether our model can be used to more deeply understand psychiatric disorders. As an initial step, we have looked at how well our model can reconstruct 200 timesteps of resting-state fMRI data in Appendix G. Furthermore, since this model is generative, we can start to probe the model and interpret the response functions when we synthetically increase communication from one region to another at certain time points. We can design communication patterns to test how communication in different regions affects the signal in others.

## 5 REPRODUCIBILITY

By making our code available after the double-blind process has concluded, we hope that our work can be reproduced and inspected by other scientists. Most of the parameters necessary for reproduction are mentioned in the experiments section, and a longer description of the formulas and math behind the model is provided in the methods section. We have explicitly added a section on the intuition behind the model so people can get a mental grasp of how the model works and that hopefully also helps people reproduce our work.

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

## A    TASK FMRI DATA RECONSTRUCTION ERROR ACROSS SPARSITY

Figure 6 shows that the Meta-CommsVAE outperforms the CommsVAE on all sparsities, but also has a clear dip around a certain sparsity. Since we do not know the ground truth of the communication in the fMRI dataset as we did with the synthetic data, the improvement in reconstruction error for higher sparsities shows promise for hyperparameter searches on other datasets. Note the reconstruction error is a squared error, summed over time and averaged over the number of regions.

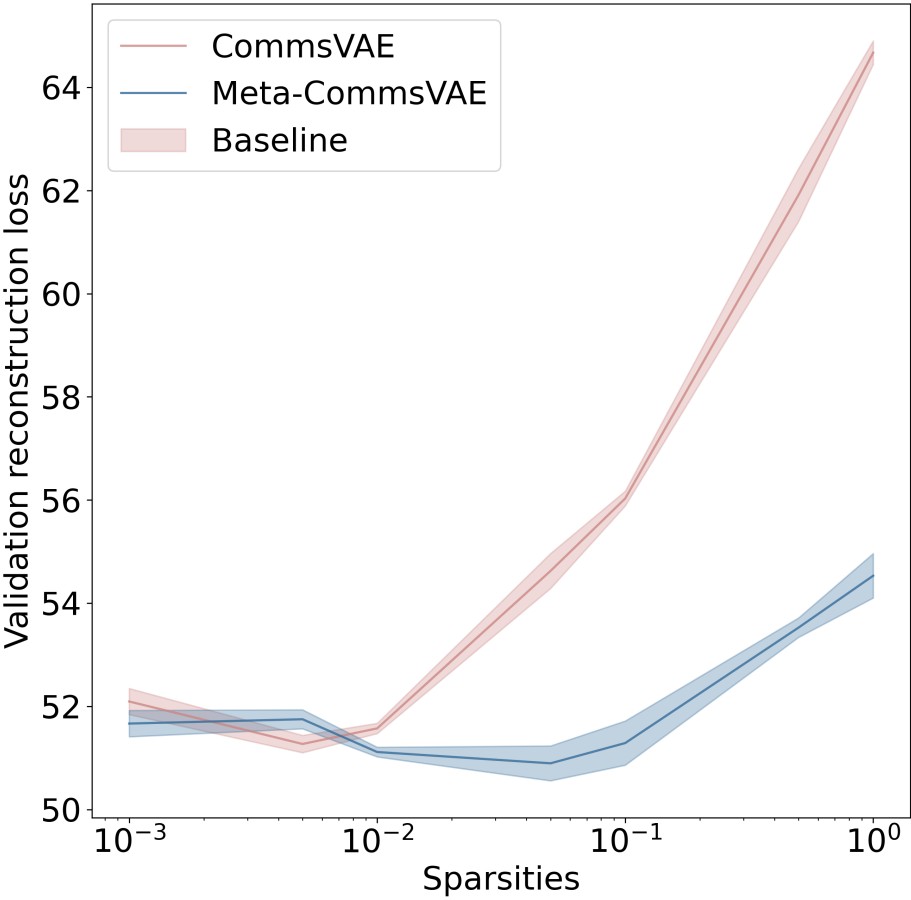

Figure 6: This figure shows the average reconstruction of both models on the fMRI dataset, with standard deviation computed across the seeds. The Meta-CommsVAE outperforms the CommsVAE for all sparsities and performs the best at a sparsity of 0.025.

# B   TASKS HEMODYNAMIC RESPONSE

Figure 7 shows the hemodynamic response for each of the tasks for the 176 timesteps we use in our model. The legend indicates what sub-task the hemodynamic response corresponds to.

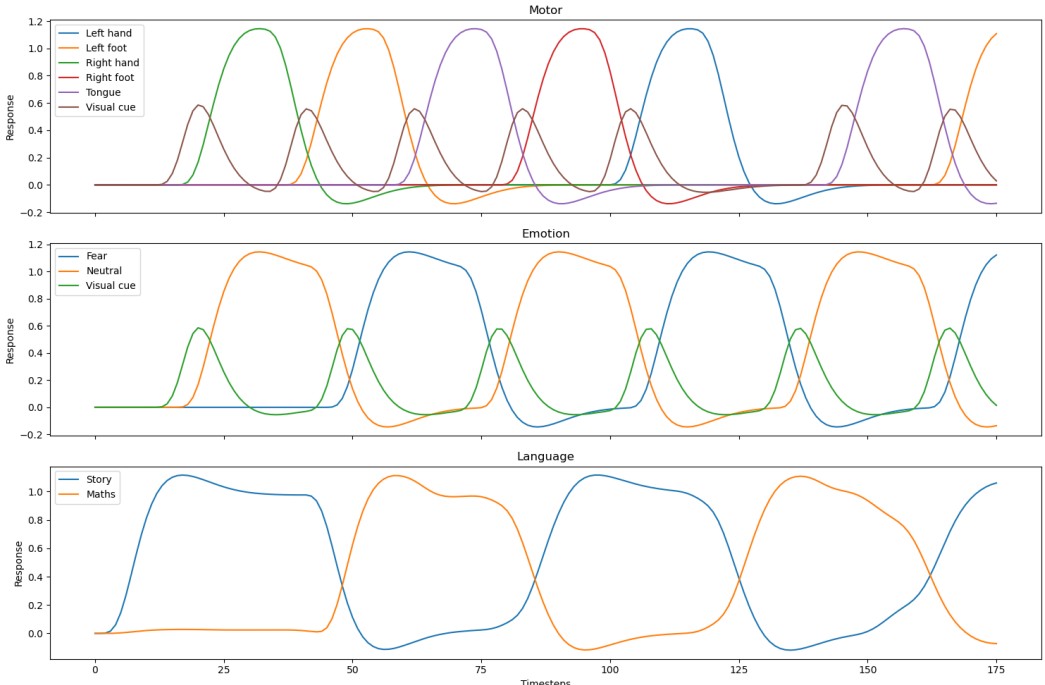

Figure 7: This figure shows the average hemodynamic response of the test set.

## C   EXAMPLE RECONSTRUCTION OF THE SYNTHETIC DATA

Figure 8 shows an example of the reconstruction of the synthetic data. The red line is the input signal, and the blue line is the reconstruction. The first row is the first region, the second row is the second region, and the last row is the third region. The columns are four random timeseries from the validation set. This shows that our model can still model the global signal, even for a shifted sinusoid, with almost perfect communication dynamics, see Figure 3,

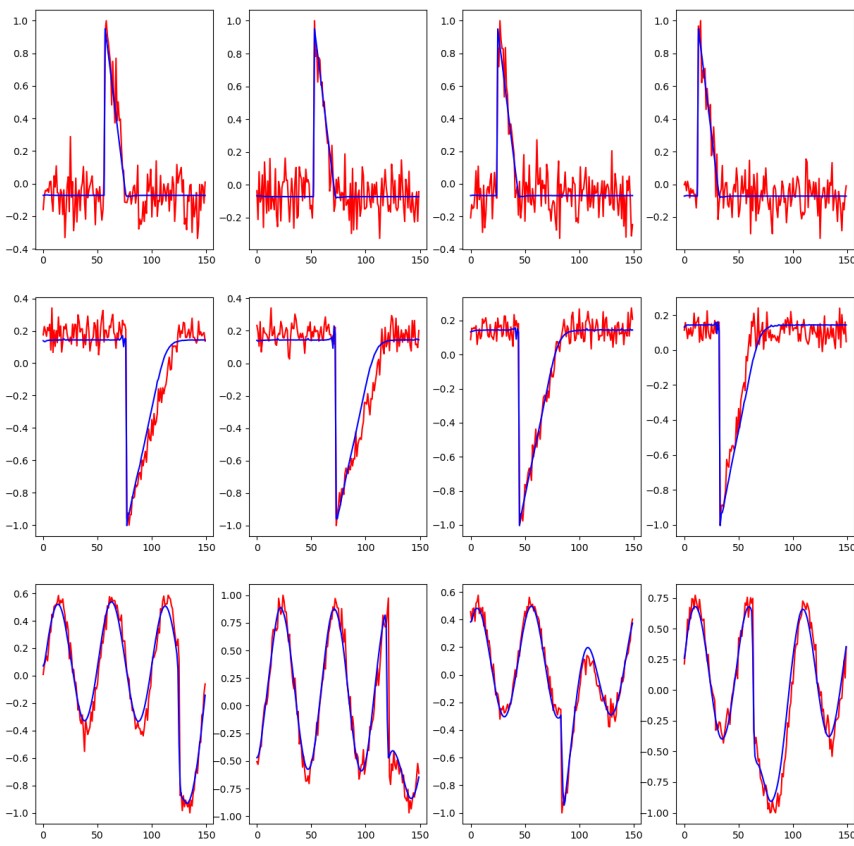

Figure 8: This figure shows an example of the reconstruction on the synthetic data for our CommsVAE model with sparsity 0.01. The red line is the input signal, the blue line the reconstructed signal, each row corresponds to a different region, and each column to a different timeseries from the validation set.

## D    SCALING

Our model in its current form is applied to data with less than twenty regions. The scaling of the model to more regions is shown in Figure 9 and indicates fairly linear scaling for the inference time, and exponential scaling in the parameters.

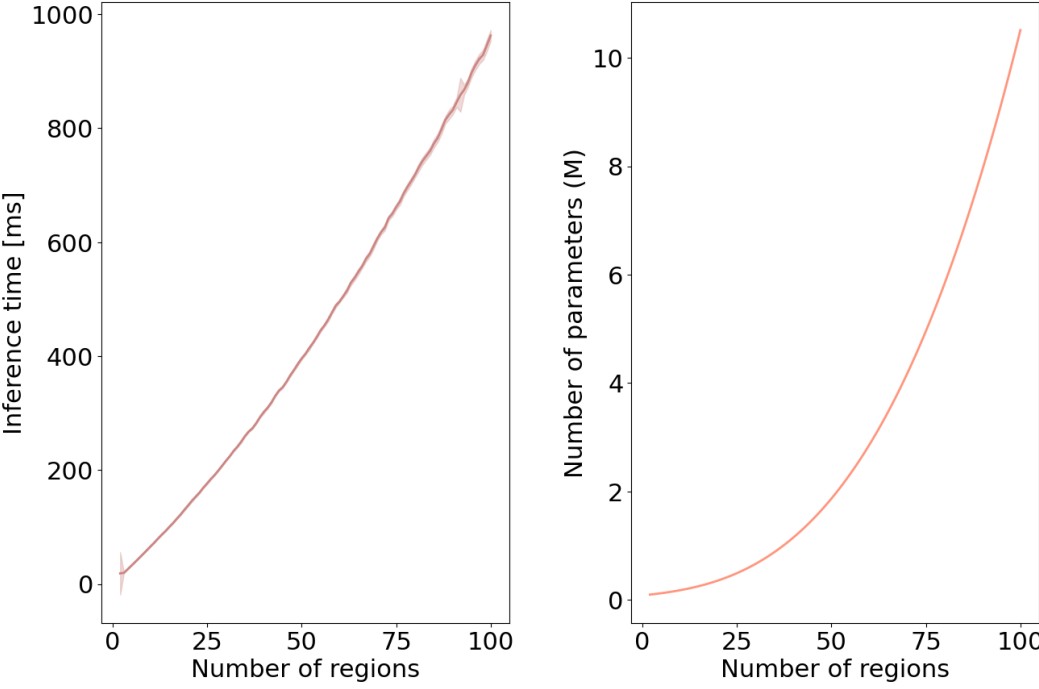

Figure 9: The scaling of the inference time in milliseconds, and the scaling of the parameters with more regions/nodes added in our network. For the inference time, the scaling seems to be largely linear, with a slight exponential component, and for the number of parameters, the scaling is exponential. The hidden sizes of the network are 16 for these calculations, similar to the ones used for the task experimentation in this work. Note that we believe that our implementation could be more efficient and potentially lead to linear scaling in the inference time.

# E    ABLATION: NO COMMUNICATION

As a way to further our understanding of the types of signal that are modeled using the communication, we compare a model trained without communication to a model trained with communication in Figure 10.

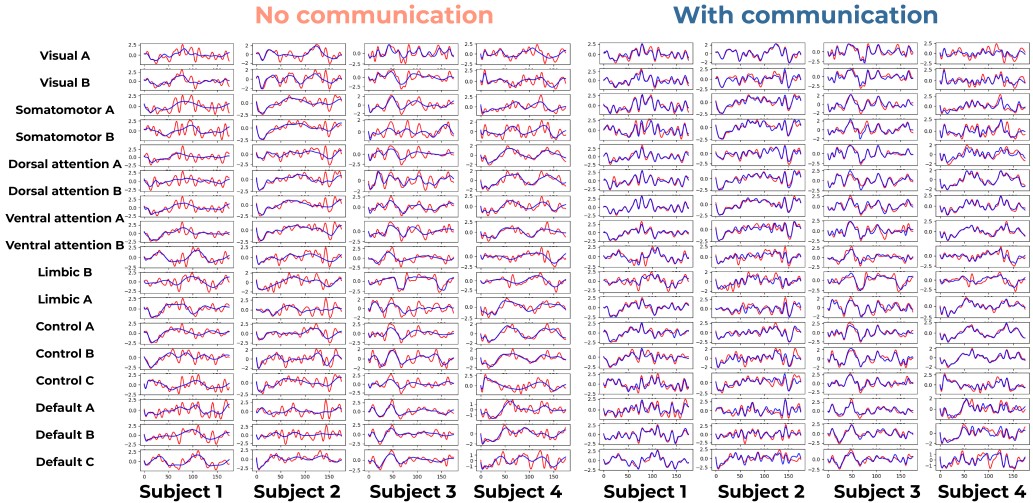

Figure 10: To indicate the difference between using communication and not using communication, we visualize the reconstruction on 4 subjects in the validation set. The left side of the figure shows a model trained without any communication, and the right side shows a model trained with communication. The red line indicates the true signal, the blue line indicates the reconstruction in both sub-figures.

# F  INDIVIDUAL TRIALS FOR SIMULATED DATA

The individual trials of our model (sparsity=0.01) on the test set. Figure 11A shows a zoomed-in version for the first 50 test trials, and Figure 11B shows the overview of all trials. Each trial exhibits the expected behavior, namely communication right at the end or atleast within the window between T1 and T2, and T2 and T3, meaning between the event in the region that is sending and the region that is receiving.

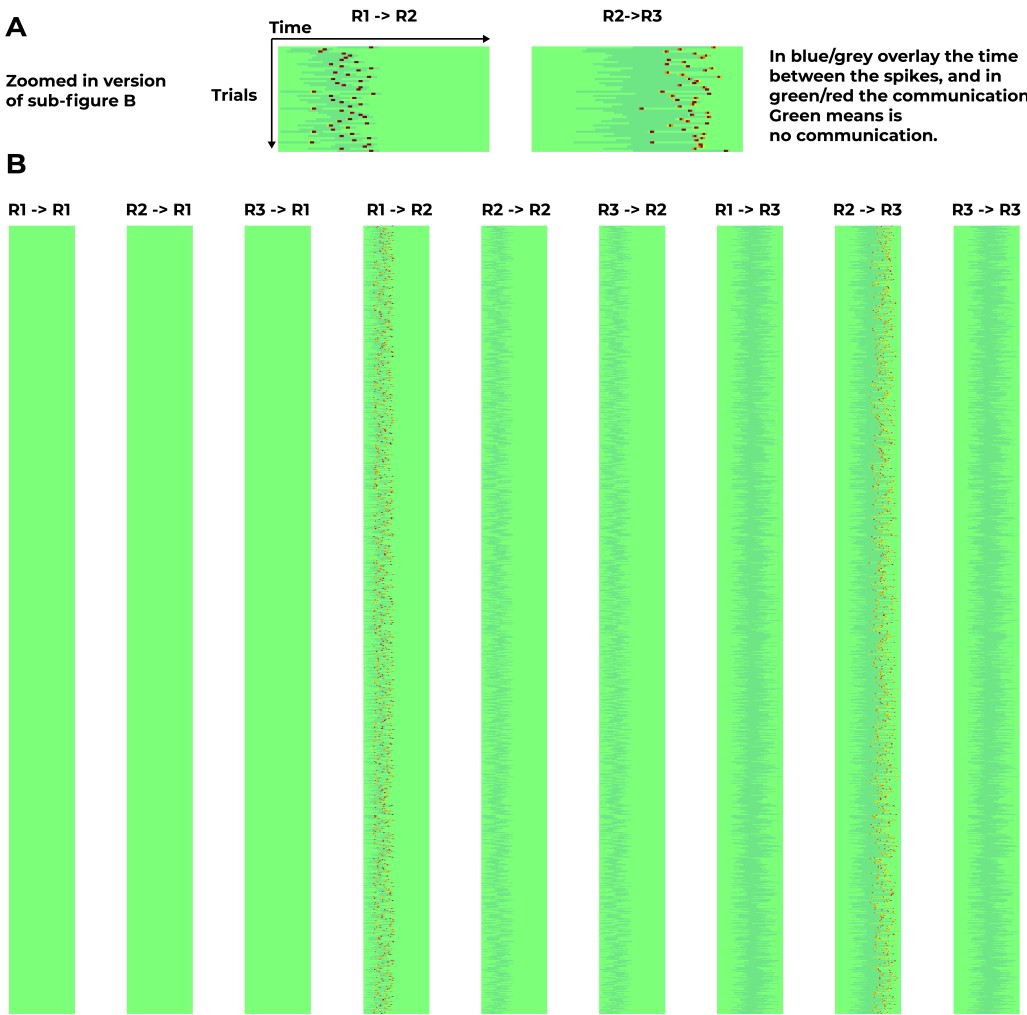

Figure 11: This figure shows the communication of the first 50 test trials in sub-figure A, and an overview of all trials in sub-figure B.

## G    RESTING-STATE FMRI

We train our model with a larger hidden size (128) on resting-state fMRI (rs-fMRI) data to ensure that sparsity also increases the reconstruction performance with higher sparsities on data that is not externally structured. We find a similar pattern as with the task data, where higher sparsities improve the reconstruction error of the model in Figure 12 and 13.

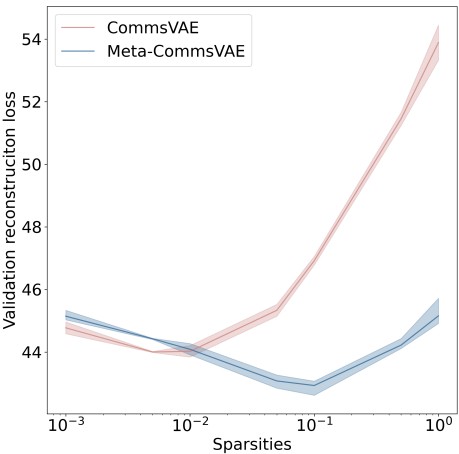

Figure 12: The validation reconstruction loss compared to sparsities for rs-fMRI data

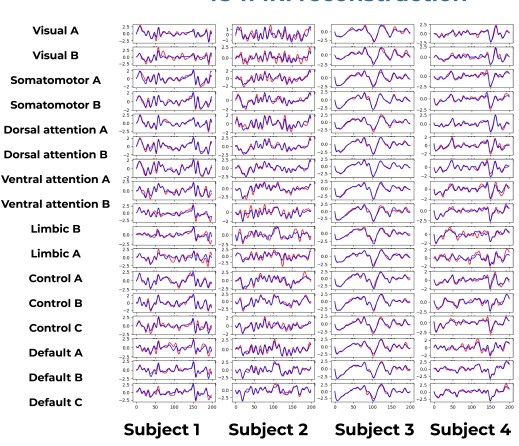

Figure 13: This figure shows the reconstruction of the rs-fMRI using our model with the best sparsity (0.1).

To make sure the communication between brain regions is sparse, especially for rs-fMRI data, we also visualize the first four brain regions for two subjects in the validation set, see Figure 14.

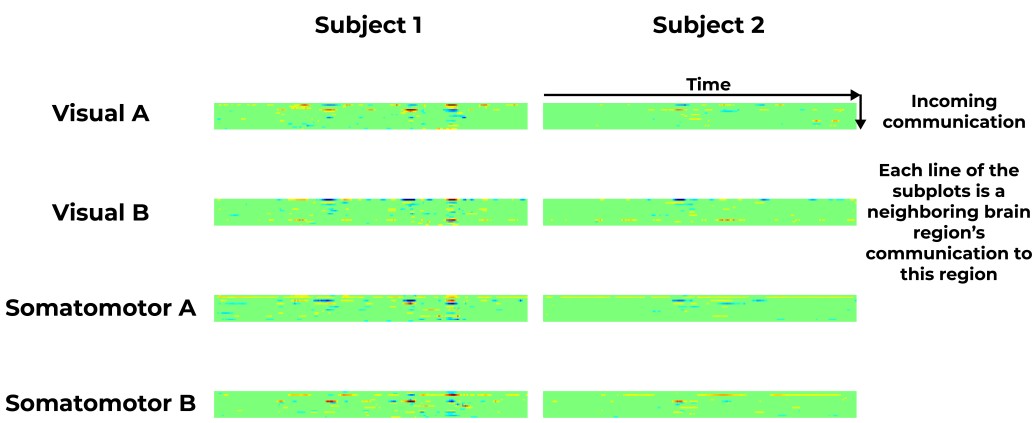

Figure 14: This figure shows the communication timeseries for the first four brain regions and two different subjects. Each line in the sub-plots refers to a brain region that is sending to the respective brain region, and the x-axis is the communication over time.

