# OpenReview forum: "CommsVAE: Learning the brain's macroscale communication dynamics using coupled sequential VAEs"
_ICLR.cc/2023/Conference — Submitted to ICLR 2023_

### Official Review · Reviewer_zo6y · 2022-10-24

**Confidence:** 4
**Clarity, Quality, Novelty And Reproducibility:** Please refer to the review section.
**Correctness:** 1
**Technical Novelty And Significance:** 1
**Empirical Novelty And Significance:** 2
**Recommendation:** 5

**Strength And Weaknesses:**

Please refer to the review section.


**Summary Of The Paper:**

In this paper, the authors propose a method for non-linear generative communication from human brain functional data, addressing three problems with connectivity approaches by explicitly modeling the directionality of communication, finding communication at each timestep, and encouraging sparsity in communication. They simulate temporal data to evaluate the proposed method that has sparse communication between embedded nodes. The proposed model can predict the expected communication dynamics. Overall, I found this paper to be difficult to follow, and I am not sure how the proposed method is novel with regard to machine learning.

**Summary Of The Review:**

a) There may be some practical applications of this paper, but it is mainly ambiguous whether its contribution(s) will be accepted as part of the machine learning area. There seems to be just a combination of some machine learning algorithms in this paper, which are then applied to a short set of data in the paper.

b) Besides the syntactic data, the only real fMRI dataset used to evaluate the proposed method is HCP-1200. For you to be able to evaluate a new approach, you will have to use data from a variety of sources.

c) I found this paper hard to follow as well. Some parts of the problem formulation are hard to understand.

d) The authors mentioned modeling the cognitive pathways from white matter — i.e.,  these regions communicate through signal propagation along the cortex and along white matter tracts over longer distances. Several theories explain fMRI may not be a good proxy to shed light on the white matter functional activities. Instead, many studies used DTI to model the regions’ communications. As an example, please see the following paper: “Learning Macroscopic Brain Connectomes via Group-Sparse Factorization”. Authors should clarify the scientific fundamental behind their proposed theory/approach.

e) The performance of the proposed method should be compared with similar state-of-the-art approaches.

f) There are some minor linguistic and typo problems in this paper.

---

> ### Author Response · Authors · 2022-11-19
> **Rebuttal reviewer zo6y**
>
> We want to thank the reviewer for their comments, and would like to invite the reviewer to clarify which parts of the paper are unclear, so we can update our manuscript and try to explain our method more in-depth.
>
> **1) There may be some practical applications of this paper, but it is mainly ambiguous whether its contributions will be accepted as part of the machine learning area. There seems to be just a combination of some machine learning algorithms in this paper, which are then applied to a short set of data in the paper.**
>
> We understand that a significant portion of our results is on neuroscience data, but we disagree with the assessment of the reviewer that this model does not apply to any other data. First, ICLR specifically invites researchers to submit research on “Applications in audio, speech, robotics, neuroscience, or any other field.” [https://iclr.cc]. Second, any temporal data that can be represented as a graph where the underlying communication between nodes is unknown, can benefit from our approach. Due to the flexibility of graph representations, this opens an enormous field of applications, such as in sociology, computational biology, and traffic prediction models. Furthermore, our method is designed based on non-linear dynamical systems theory and is an intuitive extension to latent factor dynamic modeling with additional inductive biases and increased interpretability. Lastly, to try and address your comment, we have added the following paragraph in the introduction to solidify the motivation behind our method.
> “As far as we are aware, this is the first model that explicitly models communication dynamics between vertices on a graph, specifically between brain regions. The connectivity metrics that are used currently assume that the connectivity between brain regions is stationary, e.g., Granger causality, and lack a generative model. Although non-linear generative models have been proposed for connectivity, however, determining the correct model is intractable. Furthermore, most methods quantify the coupling between brain regions and do not consider potentially rapid changes in fMRI signals that can be traced from region to region. In this paper, we specifically try to model the communication between brain regions, which likely relates to these abrupt and parsimonious changes in the signal. This means that our model is finding a type of interaction between brain regions, or more generally nodes on a graph, that has never been studied in this way before, to the best of our knowledge. The assumptions we make thus also differ from commonly used connectivity metrics. Firstly, we assume directionality and communication with the same temporal resolution as the original signal. Hence, we completely move away from windowed approaches, that quantify the coupling between brain regions within windows. Secondly, we assume that brain regions are largely independent and can be modeled by a dynamical system with sparse inputs. Lastly, we aim to learn a fully generative model of fMRI data, where the communications and initial state of each brain region's dynamical system can be sampled from a simple distribution. This also implies that our method is hard to compare with more common connectivity metrics, and we expect it to exhibit different behavior since it is modeled under different assumptions. We propose this model because we believe it has tremendous value and complements many connectivity metrics in a meaningful way, not only in the neuroimaging community, but also in other scientific fields, such as sociology, computational biology or chemistry, and traffic prediction models.”
>
> **2) Besides the synthetic data, the only real fMRI dataset used is HCP-1200.**
>
> The reason we choose HCP-1200 for this work specifically is that it is an open-source dataset so other researchers can easily replicate our work. Note though that we use three distinct tasks, each with different dynamics completely, which can be thought of as separate datasets, although still neural data. We have also added experiments on resting-state fMRI data to address this concern, which is a completely different paradigm in Appendix G.
>
> **3) I found this paper hard to follow. Some parts of the problem formulation are hard to understand.**
>
> We have rewritten the problem formulation and parts of the methodology section to motivate why we propose the model we do and hopefully improve the intuitiveness of our method. If you have any additional questions, let us know and we would be happy to answer any of them and add them to the camera-ready paper.

---

> > ### Comment · Reviewer_zo6y · 2022-12-12
> > **My score remains unchanged**
> >
> > Hello everyone. I have read all the comments and the corresponding responses. The main idea of the paper seems interesting. There are, however, a few concerns that are not adequately addressed even in the rebuttal stage, such as the evaluation of the empirical studies, the real-world application(s) of the proposed technique, and a clear presentation of the proposed approach. I still believe that further revision of this paper is necessary before it can be considered for publication. My score remains unchanged and I tend to reject this paper.

---

> ### Author Response · Authors · 2022-11-19
> **Rebuttal reviewer zo6y - continued**
>
> **4) The authors mentioned modeling the cognitive pathways from white matter. Several theories explain fMRI may not be a good proxy to shed light on the white matter functional activities. Instead, many studies use DTI to model the regions’ communications. As an example, please see the following paper: “Learning macroscopic brain connectomes via group-sparse factorization”. Authors should clarify the scientific fundamental behind their proposed theory/approach.**
>
> Although we indeed clarify in our manuscript two of the main pathways along which the brain communicates on a macro scale, we make no statement that indicates we explicitly model white matter tract activity. The statement that you reference was meant to explain that brain regions that are located further away from each other may still communicate with each other over white matter tracts. We have revised the sentence to remove any confusion about the potential implications of the sentence. To the second part of your comment, DTI cannot model functional activity over white matter tracts, since it is a static/structural data modality. In the paper you reference, the authors try to perform tractography of the white matter tracts to get an idea of the structural connectivity of the brain. The referenced paper thus has no bearing on our research, and although interesting, cannot model any of the (functional) communication dynamics that we model in our approach.
>
> **5) The performance of the proposed method should be compared with similar state-of-the-art approaches**
>
> To address this comment, we have added comparisons to Granger causality, transfer entropy, multiplication of temporal derivatives, and classification with the fMRI signal for the whole window for each brain region. Our model outperforms each of these methods convincingly. Specifically, our comparison to the classification accuracy with the fMRI signal for each brain region itself shows that our model extracts interpretable representations from the signal (communication) that lead to higher linear classification than attainable using the signal itself.
> Additionally, we have removed the more qualitative result at the end of our manuscript and replaced it with a qualitative experiment that measures the correlation between communication and sub-tasks in the data. We find that our model exhibits correlations with sub-tasks, on average over the test set and seeds, of up to 0.64.
>
>
> **6) Minor linguistic and typo problems in this paper.**
>
> We have addressed any linguistic mistakes or typos that were pointed out by other reviewers or found by us during the revision.

---

### Official Review · Reviewer_kyae · 2022-10-24

**Confidence:** 3
**Correctness:** 3
**Technical Novelty And Significance:** 3
**Empirical Novelty And Significance:** 3
**Recommendation:** 5

**Clarity, Quality, Novelty And Reproducibility:**

I found the manuscript hard to follow in sections, for instance the intuition section and the model description, where the notation can be hard to follow and may benefit from a table of parameters in the appendix. The descriptions are adequate to reproduce the results and while the code is not yet available it is slated to be posted.  I do believe the model is novel.

**Strength And Weaknesses:**

1) The model is well described and the experiments have clear descriptions for reproducibility wrt the data used and network training parameters.
2) The model explicitly incorporates the inductive bias of sparsity, and models directional binary communication between regions.
3) There are experiments with real data and synthetic data to demonstrate the model quality and types of insights it can deliver.


Weaknesses
1) Paucity of benchmarks and comparisons. I found the approach interesting, but comparing a generative model to a windowed pearson correlation felt lacking. It would be nice to see other metrics considered, especially granger causality which is often used or transfer entropy. This would be especially nice on a benchmark dataset or synthetic dataset for causal prediction, but also for the directional communication graphs in Figure 5.
In the LFADS paper that is mentioned for instance, LFADS is compared for latent factor estimation against comparable techniques in a fairly convincing synthetic data experiments.
One of the challenges in evaluating the manuscript is that while the model will certainly produce a set of communication values when applied to a dataset, it isn’t clear in what data and parameter regimes it is effective. The model will give some answer when applied to a dataset, but it isn’t clear
2) I found the description of the synthetic data could be better. How were the synthetic experiments prepared and what is the alignment metric that is used? I found the description, especially the post hoc alignment of data confusing. What are the individual data points in Figure 3?
3) In figure 4, unless I am incorrect the VAE model has been trained on the tasks before hand, so there is a greater amount of information stored in the network weights. This may contribute to accuracy classifying tasks, but also likely introduces some amount of bias.


Nits:
Reference formatting is weird – names are not separated from the text.
Figure 2 – in the cartoon there is no noise in the trace for R3, and the noise appears hand drawn (the rise in signal at T1 for R1 is not a function/does not pass the ‘vertical line test’).
Pg 4 after eq 7 typo ‘intense’
Pg 4 bottom missing equal signs
Figure 3A – what is the definition of this metric
“We train one model on all three tasks such that the prediction of the task is not influenced by the fact that the model was specifically trained to model communication for that task” – technically it is still influenced by the task, just less so.
In the introduction the authors mention an advantage over correlational approaches to function over shorter timescales, but this approach requires training data and time to initialize the state of RNNs and account for time lags, eg in the Figure 2 cartoon there is a time gap in communication that is claimed to be learned by the model.



**Summary Of The Paper:**

This work proposes new generative model of communication in graph networks, applied here to fMRI data. The network attempts to overcome weaknesses of purely correlational approaches such as pearson correlation by explicitly modeling the directional communication between different nodes in the network, and giving control of the sparsity of communication between nodes. In the model, each node possess two networks, the first a S-VAE that reconstructs the signals based on the observed data and the estimated communication and task events, and the second a network that determines whether to communicate with another node using a binary signal. This allows for the unique directional information from each node to be precisely delineated, and to regularize the sparsity of communication using a Laplacian constraint. The manuscript offers a simple example of how the model could work in practice and then applies it to an fmri dataset consisting of 3 emotional, language, and motor tasks. Because there is no ground truth for the fmri data, they look at the ability to classify tasks based on the estimated communication dynamics, and compare this to the classification ability using dFNC, a pearson correlation based approach. They find superior performance of the generative model when using shorter time windows for the dFNC approach. Lastly, they demonstrate reconstruction of a communication graph for each task for the S-VAE approach, and find some concordance with biological intuition.

**Summary Of The Review:**

I am a bit torn with this manuscript. I think that the model holds some value and has a structure (directionality, explicit communication) and inductive biases (sparsity) that could be useful , but I have reservations about the quality of presentation and comparisons given. Without benchmarks for synthetic data or better sense of how the model compares to other approaches it is unclear when it should be used. All models are wrong, some are useful, and it would be nice to know the exact circumstances and advantages this model has – in practice – compared to other approaches. I am inclined to reject unless there is significant revisions wrt benchmarks and clarity of exact situations where the model should and shouldn’t be deployed.

---

> ### Author Response · Authors · 2022-11-19
> **Rebuttal reviewer kyae - continued**
>
> **3) In what data and parameter regime is the model effective, and specifically where would this model fail?**
>
> We partly addressed this comment in the previous point, but we will reiterate and add to some of what we mentioned. An important assumption of our method is that communication is parsimonious, which we argue is biologically realistic, but can also be applied in other domains, where nodes on a graph are not constantly coupled, but switch in a non-stationary way. However, for stationary connectivity, other methods are more suited, such as Granger causality. In terms of the parameter regime, we mentioned in another comment to a reviewer that for data that is non-stationary but can be approximated well as a transient system (such as the task data in our work), it may be necessary to make sure the decoder does not become too expressive. To make these assumptions explicit in the manuscript, we have added the following paragraph:
> “As far as we are aware, this is the first model that explicitly models communication dynamics between vertices on a graph, specifically between brain regions. The connectivity metrics that are used currently assume that the connectivity between brain regions is stationary, e.g., Granger causality, and lack a generative model. Although non-linear generative models have been proposed for connectivity, however, determining the correct model is intractable. Furthermore, most methods quantify the coupling between brain regions and do not consider potentially rapid changes in fMRI signals that can be traced from region to region. In this paper, we specifically try to model the communication between brain regions, which likely relates to these abrupt and parsimonious changes in the signal. This means that our model is finding a type of interaction between brain regions, or more generally nodes on a graph, that has never been studied in this way before, to the best of our knowledge. The assumptions we make thus also differ from commonly used connectivity metrics. Firstly, we assume directionality and communication with the same temporal resolution as the original signal. Hence, we completely move away from windowed approaches, that quantify the coupling between brain regions within windows. Secondly, we assume that brain regions are largely independent and can be modeled by a dynamical system with sparse inputs. Lastly, we aim to learn a fully generative model of fMRI data, where the communications and initial state of each brain region's dynamical system can be sampled from a simple distribution. This also implies that our method is hard to compare with more common connectivity metrics, and we expect it to exhibit different behavior since it is modeled under different assumptions. We propose this model because we believe it has tremendous value and complements many connectivity metrics in a meaningful way, not only in the neuroimaging community, but also in other scientific fields, such as sociology, computational biology or chemistry, and traffic prediction models.”
>
> **4) The description of the synthetic data can be improved. How were the synthetic experiments prepared and what is the alignment metric that is used? What are the individual data points in Figure 3?**
>
> The synthetic data was z-scored before being trained on, other than that we generated independent data for training, validation, and test set. The alignment metric works as follows: ”The correspondence is calculated by convolving a normal distribution over an impulse response centered on the start of the event in that brain region. Then we correlate the communication between the resulting signal and the communication to that brain region for different shifts of the communication signal. Finally, we sum the maximum absolute correlations for R1->R2 and R2->R3 and subtract the sum of two times the average absolute communication for all other brain regions. This corresponds to expecting zero communication in all other brain regions, and a spike right before the start of the actual event. By using shifts to calculate the correlation we do not make assumptions about the lag that is learned between the communication RNN and the decoder.” The individual data points in the plot are different seeds the model is trained with. We have added clarifications concerning the reviewer’s questions in the revised manuscript as well.

---

> ### Author Response · Authors · 2022-11-19
> **Rebuttal reviewer kyae - continued - part two**
>
> **5) The VAE model has been trained on the tasks beforehand, this could introduce some amount of bias.**
>
> leaked into the communication signal (although it is only 1-dimensional) for the subjects in the training set. However, we have updated our model to address this comment and the comment from another reviewer, by adding a threshold (0.1) during training for the mean and the samples from the communication distributions. This threshold is equal to one standard deviation of the prior, and although some samples may still be able to fall outside of this threshold during training (even if the mean of the distribution is zero), we use the mean of the distribution during inference. This effectively results in a sparse inference signal. Additionally, although there may be noise in the hidden states regarding the subjects in the training set, we test our predictions based on subjects in the test set. We would expect the generalization of the model to significantly decline in case of clear noise, and this does not happen.
> It should also be noted that there is previous work that shows that unsupervised training works as an “unusual form of regularization” [1]. This means that our training method helps balance the model between variance and bias [2]. And, because it is an autoencoder (i.e., uses a reconstruction loss), it does not add new information or bias that cannot be found within the data itself. It is also a fairly well-described phenomenon that autoencoder latent spaces reduce noise.
>
> [Citation 1 [1]](https://www.jmlr.org/papers/volume11/erhan10a/erhan10a.pdf)
> [Citation 2 [2]](https://link.springer.com/article/10.1057/jors.1993.11)
>
> **6) Reference formatting is weird.**
>
> We have updated the references in the updated manuscript.
>
> **7) The noise in Figure 2 seems hand drawn.**
>
> We have updated the description of Figure 2 to reflect that Figures 2A and B are drawn in graphical design software.
>
> **8) Minor grammatical and spelling errors. Clarity of the model description.**
>
> We have updated the manuscript to remove the grammatical and spelling errors the reviewer points out. Thank you for being so kind as to specifically point them out! We have also updated our manuscript to try and improve the clarity of our methods section based on the reviewer's comments.
>
> **9) This approach still requires training data and time to initialize the RNNs and account for time lags. E.g., in Figure 2 there is a time gap between the communication that is claimed to be learned by the model.**
>
> We agree that our approach requires training data and time to train the model. However, we argue that this is the case for many approaches. Performing data discovery with simple methods is always preferred over training neural networks on data right away, this could lead to all sorts of downstream issues, such as bias. However, our method can extract a specific type of information from the data, namely communication, and is the first method to do so as far as we are aware. We also show that communication leads to better task discriminability but is likely to have a significant impact on our understanding of mental disorders, especially considering the easy extensibility to rs-fMRI data, see Appendix G. To the second part of the comment, the time lag is not a hyperparameter in our model and our model can comfortably learn any time lag in the signal as shown in our synthetic data results.

---

### Official Review · Reviewer_sVVG · 2022-10-26

**Confidence:** 3
**Correctness:** 2
**Technical Novelty And Significance:** 4
**Empirical Novelty And Significance:** 3
**Recommendation:** 5

**Clarity, Quality, Novelty And Reproducibility:**

Writing is clear.
Novel neural modeling architecture, but the analyses are not too convincing.

**Strength And Weaknesses:**

This paper seems like a reasonable extension of the popular neural modeling approaches (ex. LFADS) to fMRI and multiple brain regions.

I have some reservations:
1. I am not confident on the identifiablity of the model. If the generator for each region is expressive enough, it might not be necessary to have any communication from other brain regions.

2. I am not confident of the instantaneous sparsity as biologically realistic. If region A is not connected with region B, then the communication will be 0 for all time points. This points to some sort of group sparsity over time/trials.

3. Do you recover the ground-truth communication on individual trials? In simulations (Fig 3), all results are averaged across multiple trials. In my experience, it is hard to get a truly sparse signal as posterior for individual trials in VAE setting. The authors should clarify whether the sampled posterior is sparse or not.

4. The fMRI analyses seem weak. The biological insights seem mixed and not too clear.

**Summary Of The Paper:**

This paper models the fMRI activity of multiple brain regions simultaneously, with sparse communication between them.

**Summary Of The Review:**

More analyses on identifiability, or more empirical analysis would be helpful. At the current state, the paper seems more suited for a neuroscience journal after using it for scientific insights.

---

### Official Review · Reviewer_qncm · 2022-11-01

**Confidence:** 5
**Correctness:** 3
**Technical Novelty And Significance:** 2
**Empirical Novelty And Significance:** 3
**Recommendation:** 6

**Clarity, Quality, Novelty And Reproducibility:**

Clarity: While overall, the manuscript is easy to read and the figures (especially Fig. 1) help a lot in understanding the design of the model, the methodology section can be improved for its organization. Specifically, the section “Generative communication models” is more like an introduction. Further, the paragraph led by the sentence “One recent model that has been used to model this dynamical system is LFADs…” mixed the S-VAE and RNN+MLP models together, making it difficult to understand each component clearly. Finally, equations 8-9 can use more explanations for the symbols used and their relationships to the corresponding models.

Quality: The quality of the manuscript is good, and the results are well-supporting the authors’ claims.

Reproducibility: In addition to the public source code promised by the authors, the proposed model shall be reproducible based on the description in the methodology and the Implementation sections.

**Strength And Weaknesses:**

Strength: Explicit modeling of the inter-region communication is both interesting and important for the fMRI data analysis and, more generally, the majority of multi-variate time series data analysis. The design of the model by S-VAE and RNN is reasonable and well-implemented.

Weakness: First of all, it will be important to compare the proposed model with the “traditional deep learning” approaches. Specifically, it will be interesting to see ablation experiments of 1) training an S-VAE using the signals from all regions as input and output (with certain regularizations) and 2) removing the communication model to train the individual S-VAEs only. The reviewer is also curious about how the proposed model would scale to the region (node) size, as the current experiment on real data includes a relatively small (<20) number of regions. This concern is especially prominent as 1) the manuscript did not provide computational cost for training the model, and 2) each node will have its own S-VAE and RNN+MLP to be trained. The final concern is more of a naming convention issue, as generally, the communication model developed in this work is not considered “Generative.” Rather, it performs a binary classification on whether a specific inter-region communication will happen at a specific time point. The methodology section of this manuscript also suffers from clarity issues; see comments below.

**Summary Of The Paper:**

The manuscript proposed a Generative Communication Model by individually modeling the signals (by an S-VAE) and the communication sent at each graph node (by an RNN+MLP). Experiment on synthetic data shows that the model can successfully recover the designed communication between regions. Experiment on real (HCP) fMRI data shows good classification accuracy of the task type, and meaningful inter-region communication patterns can be discovered.

**Summary Of The Review:**

An interesting and important attempt to model the connectivity of fMRI/time series data via explicit modeling of the inter-region communication. More investigations into how and why the model can work will be needed.

---

### Author Response · Authors · 2022-11-19
**General rebuttal**

We want to thank all of the reviewers, and appreciate that R1, R2, and R3 found the idea and methodology novel and interesting, and hope to have addressed  your concerns regarding the experimentation by adding additional comparisons, training the model on resting-state fMRI data, and replacing the qualitative results with a comparison to the ground truth sub-tasks in the data.  We have made some small changes to the model, namely the addition of a threshold (+-0.1) during training on both the samples from the communication distribution(s), and the mean of the distributions themselves. We did not find any large changes in the results, but we now have certainty that the communication during inference is sparse. Furthermore, we made a slight change that also did not greatly affect the results, but improves the biological plausibility, and theoretical backing of the model. Namely, instead of receiving communication **c** at time **t** from region **i** at time **t**, the receiving region will receive communication that was sent one timestep before. This means that the model is temporally causal instead of temporally non-causal.

If the reviewers have any additional concerns, we will be happy to address them or clarify them in the upcoming period.

---

### Author Response · Authors · 2022-11-29
**We would appreciate additional feedback!**

Dear Reviewers,

We hope our revision increases your confidence that our work is novel and does in fact produce important representations that other methods are not able to find. We highly appreciate your initial reviews, and would love to know what you think of our revisions.

---

### Decision · Program_Chairs · 2023-01-20

**Decision:**

Reject

**Justification For Why Not Higher Score:**

Lack of utility and validation.

**Justification For Why Not Lower Score:**

N/A

**Metareview: Summary, Strengths And Weaknesses:**

The paper proposes modeling fMRI measurements with latent dynamical system expressed in terms of a macroscale communication dynamics using coupled sequential VAEs. The latent structure of the communication is then interpreted. The reviewers found the premise that causal communication links could be inferred from extremely coarse (poor spatial and temporal resolution), and purely observational measurements such as fMRI to be implausible. Further, no evidence is given to validate the predictions of the model. The reviewers found the model to be of questionable utility to both neuroscientists and machine learners.

**Summary Of Ac-Reviewer Meeting:**

The reviewers were in unanimous agreement that they could not understand the utility of the proposed model. They were not convinced that the method (commsVAE) and the data (fMRI) were suitable, even in principle, for estimating the connectivity between brain regions. Further, since no validation of the predictions is provided, it is hard to judge the accuracy. Finally, it was not clear if there was any other way of interpreting the structure of connectivity discovered. During the discussion all reviewers unanimously advocated for rejection.